# Overcoming Vemurafenib Resistance in Metastatic Melanoma: Targeting Integrins to Improve Treatment Efficacy

**DOI:** 10.3390/ijms25147946

**Published:** 2024-07-20

**Authors:** Asiye Busra Boz Er, Helen M. Sheldrake, Mark Sutherland

**Affiliations:** 1Institute of Cancer Therapeutics, School of Pharmacy and Medical Sciences, University of Bradford, Bradford BD7 1DP, UK; asiyebusra.bozer@erdogan.edu.tr; 2School of Chemistry and Biosciences, University of Bradford, Bradford BD7 1DP, UK

**Keywords:** BRAF, drug resistance, integrin, melanoma

## Abstract

Metastatic melanoma, a deadly form of skin cancer, often develops resistance to the BRAF inhibitor drug vemurafenib, highlighting the need for understanding the underlying mechanisms of resistance and exploring potential therapeutic strategies targeting integrins and TGF-β signalling. In this study, the role of integrins and TGF-β signalling in vemurafenib resistance in melanoma was investigated, and the potential of combining vemurafenib with cilengitide as a therapeutic strategy was investigated. In this study, it was found that the transcription of *PAI1* and *p21* was induced by acquired vemurafenib resistance, and *ITGA5* levels were increased as a result of this resistance. The transcription of *ITGA5* was mediated by the TGF-β pathway in the development of vemurafenib resistance. A synergistic effect on the proliferation of vemurafenib-resistant melanoma cells was observed with the combination therapy of vemurafenib and cilengitide. Additionally, this combination therapy significantly decreased invasion and colony formation in these resistant cells. In conclusion, it is suggested that targeting integrins and TGF-β signalling, specifically *ITGA5*, *ITGB3*, *PAI1*, and *p21*, may offer promising approaches to overcoming vemurafenib resistance, thereby improving outcomes for metastatic melanoma patients.

## 1. Introduction

Metastatic melanoma is a deadly form of skin cancer with increasing incidence rates. Vemurafenib, a BRAF inhibitor drug approved by the FDA, has shown promise in the treatment of BRAFV600E mutant melanoma [1]. However, the development of resistance to vemurafenib within a short timeframe leads to poor prognosis for patients. To improve patient outcomes, an increasing body of research has focused on drug resistance [2].

Integrins are cell surface receptors that mediate cell–cell and cell–matrix interactions, playing crucial roles in signalling pathways relevant to cancer development and progression. Significant attention has been directed towards the role of integrins in promoting tumour progression and drug resistance [3]. For example, forced expression of integrin α5β1 is associated with doxorubicin resistance in MCF7 breast cancer cells [4]. Seguin et al. demonstrated that integrin αvβ3 is overexpressed in lung and breast cancer cells and interacts with KRAS and RALB to drive erlotinib resistance [5]. These findings collectively highlight the significance of RGD-binding integrins in drug resistance across different cancer types. The mechanism by which integrins mediate drug resistance is complex and multifaceted. Integrins play crucial roles in cell adhesion, migration, and signalling pathways that influence cancer cell survival and response to therapies. They contribute to the activation of key signalling pathways, such as MAPK and PI3K/AKT, which promote cell survival and resistance to therapy. Integrin-mediated signalling can also impact the tumour microenvironment and promote tumour progression, angiogenesis, and metastasis. These effects on the tumour microenvironment can further influence the efficacy of therapeutic interventions. For example, the activation of MAPK and PI3K/Akt signalling by hepatocyte growth factor [6], or other components present in serum [7], has been shown to mediate a mechanism of intrinsic tumour resistance to BRAFV600E inhibition. Integrin-mediated adhesion to fibronectin protects cells from apoptosis induced by the deletion of *BRAF* or *Mcl-1* [8,9]. Activation of stromal fibroblasts by the BRAFV600E inhibitor PLX4720 promotes resistance by fibronectin matrix synthesis and β1 integrin signalling [10], and a similar *α*5β1-dependent resistance pathway has also been observed in melanoma cells [11].

Targeting integrins, either through small molecules or antibodies, has been explored as a potential approach to overcome drug resistance in various cancers. Inhibitors of integrins αvβ3 and α5β1, such as cilengitide, have shown promising results in preclinical studies. β3 integrin interacting with KRAS has been shown to promote resistance to EGFR inhibition, which is reversed by inhibiting β3 signalling [5]. Resistance to other targeted agents is also reversed by β3 integrin antagonists or knockdown [12,13]. The combination of an anti-*α*v integrin antibody with Src kinase inhibition increased anti-proliferative and anti-migratory activity in colon cancer cell lines [14]. This increased anti-migratory activity in Src-inhibitor-resistant cells suggests that integrin/kinase combination therapy will be valuable as an anti-metastatic strategy regardless of a tumour’s intrinsic or acquired resistance to kinase inhibition.

β_3_ expression is increased in A375 cells treated with PLX4720, and further increased in an A375 acquired resistance line. β3 siRNA knockdown or an anti-αv integrin antibody were shown to increase response to PLX4720 in a B16-V600E acquired resistance model [15].

BRAF signalling and the downstream effects on PD-L1 have been shown to be controlled by αvβ3 signalling [16]. Cilengitide treatment in A375 human and B16 mouse melanoma cells significantly decreased colony formation and induced apoptosis [17]. Cilengitide decreased the expression of PDL-1 through STAT3 downregulation in a B16 mouse melanoma. Combination with anti-PD1 maximised the benefits of PD-1 therapy and reduced tumour growth and prolonged survival by activating CD8+T cells in a murine melanoma model [17].

Integrin expressions are variable in cancers, and can also change during tumour progression and in response to drug treatment [3]. This situation makes it hard to address a specific integrin to target [18]. The constitutive high expression of αvβ3 in melanoma and its interplay with BRAF signalling render it an attractive starting point for investigating combination therapy. In this work, we aim to demonstrate the potential of integrin–BRAFV600E inhibitor combinations, considering both αvβ3 and identifying novel integrin targets based on their changes in response to the development of resistance to BRAFV600E inhibitor treatment. The data generated could be extended to the clinic to potentially improve treatment response and overcome resistance.

## 2. Results

### 2.1. BRAF V600E Mutant Cell Lines Are More Sensitive to Vemurafenib

To understand BRAFV600E inhibitor response in melanoma when generating BRAFV600E-inhibitor-resistant cell lines, we first compared the vemurafenib sensitivity of BRAFV600E mutation carrying the melanoma cells lines UACC62, A375, SK-MEL28, and M14 with BRAF WT carrying the melanoma cells lines MeWo and SK-Mel2. BRAF WT melanoma cells displayed higher viability than BRAFV600E mutant melanoma cells when treated with vemurafenib. While the IC_50_ values of vemurafenib in the BRAFV600E mutant melanoma cells were less than 0.5 μM, the BRAF WT melanoma cells had an IC_50_ of more than 20 μM (Table 1).

### 2.2. Generation and Morphology of Resistant Cells

The BRAFV600E mutant cell lines A375, M14, SK-MEL28, and UACC62 were made resistant to vemurafenib by gradually increasing doses (0.1–10 μM) over a period of 4 months. To confirm the newly generated cells were resistant, MTT viability assays were performed and the IC_50_ results of the cells were analysed. Chronic vemurafenib exposure resulted in gain of resistance and cells exhibited higher IC_50_ levels. Overall, A375’s IC_50_ increased approximately 22-fold, M14’s IC_50_ increased approximately 12-fold, SK-MEL28’s IC_50_ increased approximately 66-fold, and UACC62’s IC_50_ increased approximately 86-fold between parental and resistant cells (Table 2).

To examine the cell morphology, A375, M14, SK-MEL28, and UACC62 cells were seeded as 30% confluent and incubated for 24 h without vemurafenib to avoid drug stress on the cells. Under light microscopy, the resistant cells displayed more polarised and longer morphologies when compared with the parental cells. The UACC62 and SK-MEL28 cells became larger and more polarised (Figure 1A). The A375 cells gained longer and spindle-like features (Figure 1A). The M14 cells gained more mesenchymal morphologies, becoming bigger and longer but not rounded in shape (Figure 1A). This suggests that acquiring resistance may induce some changes in protein expressions which affect the morphology of the cells. The difference between the parental and resistant cell lines was compared using ImageJ 1.53e, and a significant increase in the ratio of the major to minor axis between the parental and resistant A375, SK-MEL 28, M14, and UACC62 cells was observed (Figure 1B).

### 2.3. Vemurafenib Resistance Induces ITGA5 Transcription in Melanoma Cells

Fibrinogen is one of the ligands of αvβ3, α5β1, and αIIβ3 RGD-binding integrins, but how it affects their transcription levels is unknown. However it is known that the presence of fibrinogen can induce integrin-associated pathways [19], so before using fibrinogen in further experiments as a model of ECM binding, its effects on integrin transcription levels were analysed. To determine whether fibrinogen induced integrin expression, A375 and SK-MEL 28 cells were seeded in plates either coated or uncoated with fibrinogen, and the cells were collected at various time points. The largest response to fibrinogen was observed at 4 h for A375 and 6 h for SK-MEL 28; therefore, in subsequent experiments, these time points were used for collecting and analysing the cells.

To determine the integrins potentially involved in vemurafenib acquired resistance, the gene expressions of *ITGB1*, *ITGB3*, *ITGB5*, *ITGB6*, *ITGB8*, *ITGA2*, *ITGAIIB*, *ITGA4*, *ITGA5*, and *ITGAV* were compared for the parental (P) and resistant (R) A375 and SK-MEL28 cells in the presence and absence of fibrinogen by real-time PCR. A statistically significant increase was observed in the *ITGA5* levels for both cell lines. *ITGA5* levels increased 2.3-fold in A375R cells and 9-fold in SK-MEL28R cells (Figure 2A–C). An increase was observed in *ITGB1* and *ITGB3* in parental cells in the presence of fibrinogen in both A375P and SK-MEL28P cells. Also, an increase was observed in *ITGB6* in SK-MEL28P cells in the presence of fibrinogen (Figure 2B), but a decrease was observed in resistant cells with fibrinogen, and no change was observed between the parental and resistant cells. Some genes had no response in different cell lines. For SK-MEL28R cells, *ITGA2* gene expression, and for A375R cells, *ITGB5*, *ITGB6*, *ITGA4*, and *ITGAV* gene expression was unchanged in the resistant cells and parental cells in the presence of fibrinogen (Figure 2B). Also, *ITGB8* and *ITGA4* gene levels were significantly decreased in SK-MEL28R cells (Figure 2A,B), but not in A375R cells.

### 2.4. Vemurafenib Resistance Induces β3 and α5 Expression

Gene expression levels and protein levels were not strictly correlated. To understand the correlation between gene and protein expression, integrin β3 and α5 subunit protein levels were analysed by Western blotting. A significant increase in protein expression between parental and resistant cells was observed for both A375 and SK-MEL 28 cells (Figure 3A,B). For the A375 cell line, an approximately 15-fold increase in integrin β3 and a 1.8-fold increase in α5, and, for the SK-MEL28 cell line, an approximately 2-fold increase in β3 and a 5.5-fold increase in integrin α5 levels was observed in the resistant cell lines (Figure 3B).

The localisation of β3 and α5 proteins were analysed by immunofluorescence. In the A375P cells, integrin β3 was localised in both the nucleus and cytoplasm, but there was also a noticeable localisation in the nuclear membrane (Figure 3C—red arrows). In the A375R cells, integrin β3 was localised both in the nucleus and cytoplasm, but was not noticeable in the nuclear membrane, which is possibly related to the condensed rounded shape of the resistant cells. In the A375P cells in the absence of fibrinogen, integrin α5 was also localised both in the cytoplasm and nucleus. In the presence of fibrinogen, cells looked more attached to the surface, but integrin α5 was still located in both the nucleus and cytoplasm. In the A375R cells, in the absence and presence of fibrinogen, integrin α5 was localised both in the cytoplasm and nucleus, but, interestingly, also in dot-like structures around the cells (Figure 3D—red arrows).

### 2.5. Vemurafenib+Cilengitide Combinations Have a Synergistic Effect on Vemurafenib-Resistant Melanoma Cells

The MTT assay was used to determine the sensitivity of the A375P, A375R, SK-MEL28P, and SK-MEL28R cells to the integrin inhibitor cilengitide. The results showed that resistance to vemurafenib did not affect the cells’ response to cilengitide (Table 3).

To analyse the combination therapy, five concentration points at a 1:1 ratio for each drug were used according to the Chou and Talalay method [20]. The combination effect was analysed by calculating the combination index (CI) based on the median effect principle. The colour and the type of the line in the polygonogram indicate the type of interaction. The results show that the vemurafenib+cilengitide 1:1 drug combination is moderately antagonistic in the A375 and SK-MEL28 cell lines, very strongly synergistic in the A375R cell line, and synergistic in the SK-MEL28R cell line (Figure 4, Table 4).

### 2.6. Combination Therapy Decreases Invasion and Colony Formation

To observe the effects of the combination therapy on invasion, spheroids were prepared from resistant cells and embedded into Matrigel and treated with vemurafenib, cilengitide, or vemurafenib+cilengitide at the IC_50_ concentrations of each drug.

SK-MEL28R cells formed proper rounded spheroid structures (Figure 5A–D) and, at the end of the incubation in Matrigel with vemurafenib and cilengitide monotherapies and combination therapy, significant changes were observed in the invasion behaviour. Spheroids treated with vemurafenib monotherapy showed increased invasion compared to the control but, in the vemurafenib+cilengitide combination, invasion decreased more than 5-fold compared to the control (Figure 5E).

The colony-forming capability of cells is directly linked with their tumourigenic features, such as survival and tumour-initiating capabilities. Using a low density of cells and analysing their capability to survive and form colonies under different treatments has been used to analyse cell response and behaviour to certain treatments [21]. To analyse the effect of combination therapy on the capability of colony formation, SK-MEL28 parental and resistant cells were seeded on fibrinogen-coated and non-coated flasks, treated with vemurafenib and cilengitide as monotherapy and combination therapy for 11 days, and colonies were counted after crystal violet staining.

SK-MEL28 parental cells formed more colonies in fibrinogen-coated flasks than uncoated flasks (Figure 6A,B). For both fibrinogen-coated and non-coated conditions, a decrease in colonies was observed between the control and treatments. Cilengitide monotherapy gave a greater decrease in colony formation than vemurafenib monotherapy. The vemurafenib+cilengitide combination significantly decreased the colony formation compared to either drug alone (Figure 6B).

Similar to the SK-MEL28 parental cells, SK-MEL28-resistant cells also formed more colonies in fibrinogen-coated plates (Figure 6A,B). In fibrinogen-coated wells, cells treated with vemurafenib exhibited an increase in colony formation; this was not observed in the non-coated flasks. Cilengitide alone significantly decreased the colony formation and the combination of vemurafenib+cilengitide totally inhibited colony formation (Figure 6B).

### 2.7. Vemurafenib Resistance Induces the MAPK Pathway in Melanoma Cells

It is known that the MAPK pathway reactivates when cells gain resistance to vemurafenib [22], but the association with integrins is still unknown. To confirm the MAPK signalling increase and the effect of fibrinogen on MAPK, the protein expressions of pERK1 and pERK2 were investigated in A375P and A375R cells. Cells were grown without FBS in media for 24 h, and then seeded into 10 cm fibrinogen-coated and non-coated plates and incubated for 4 h. The expression levels were normalised and quantitated with the β-actin levels (Figure 7A,B).

Phosphorylated elements of MAPK showed that the MAPK pathway is activated both in resistant cells and parental cells in the presence of fibrinogen. An approximately 8-fold increase in pERK1 levels and an 8-fold increase in pERK 2 levels was observed between A375 parental and resistant cells. Also, in the presence of fibrinogen, a 4-fold increase for pERK1 and a 4-fold increase for pERK2 was observed in A375 parental cells. There was no significant change observed between the resistant cells in the presence of fibrinogen (Figure 7A,B).

ERK1/2 localisation was analysed in A375P and A375R cells in the presence and absence of fibrinogen by immunofluorescence. ERK1/2 was localised in the cytoplasm in parental A375 cells (Figure 7C). ERK1/2 translocated to the nucleus and was localised in both the nucleus and cytoplasm in resistant cells grown on fibrinogen. When resistant cells were grown on fibrinogen, this increased ERK protein accumulation in the nucleus compared to resistant cells grown in the absence of fibrinogen.

### 2.8. Vemurafenib Resistance Induces TGF-β in Melanoma Cells

To determine active integrin-related pathways, parental and resistant A375 and SK-MEL28 cells were grown on fibrinogen-coated and non-coated plates, and gene expression levels were compared for the targets *MAPK1, MAP2K1, MAPK3, RELA, RELB, p21, IKKα, HRAS, SMAD4, SMAD7, AKT1, PAI1, PDL1,* and *PD-1*.

A significant decrease was observed in *MAPK1*, *MAP2K1*, *IKKα*, *HRAS*, *SMAD4*, and *AKT1* levels between the parental and resistant SK-MEL28 cells (Figure 8B). Also, a significant increase was observed in *PAI1* and *p21* levels in both SK-MEL28- and A375-resistant cells (Figure 8C,D).

To further investigate the TGF-β activity in parental and resistant cells and in the presence of fibrinogen, the TβRI kinase-deficient receptor (TGBR1-K232R), TβR1-TD (TGBR1-T204D) constitutively active receptor, and TβR-II receptors were overexpressed and compared with the mock cells (cells transfected with empty vector) by using 3TP-LUX, SBE-LUC, and Pare-LUX.

In A375 and SK-MEL28 parental cells, a slight TGF-β activation was observed from the overexpression of TβRI-TD and TβR-II receptors; however, significantly increased TGF-β activity was observed in resistant cells (Figure 9A–F). The overexpression of these plasmids can be counted as a positive control. On the other hand, the overexpression of TβRI-KR, which decreases signalling at the receptor level, significantly decreased the TGF-β response. In mock-transfected resistant cells, significant TGF-β activation was observed for the three TGF-β responsive elements, 3TP-LUX, SBE-LUC, and PARE-LUX, and the level of activation was similar to the positive controls both in SK-MEL28 and A375 cells, which supports vemurafenib resistance increasing TGF-β signalling (Figure 9A–F). Furthermore, in the presence of fibrinogen, resistant cells showed a cumulative increase in the activity of TGF-β, not only in TβRI-TD- and TβRII-transfected cells, but also in mock-transfected cells, which indicates that TGF-β signalling is mediated by fibrinogen binding a receptor. Observing the same response in both resistant melanoma cell lines suggests that TGF-β is a key actor for vemurafenib drug resistance.

To understand TGF-β’s effect on the expression of TGF-β-responsive genes and *ITGA5* in parental and resistant cells, the TβRI-KR kinase-deficient receptor (TGBR1K232R), which can localise to the cell membrane and prevent signal transmission to suppress TGF-β’s effects in cells, was overexpressed in the presence and absence of fibrinogen.

A sharp increase in *PAI1*, *p21*, and *ITGA5* levels was observed between parental and resistant cells (Figure 9G–I). An 80-fold increase in *PAI1* (Figure 9G), a 22-fold increase in *p21* (Figure 9H), and a 17-fold increase in *ITGA5* (Figure 9I) was observed in resistant cells. When TGF-β signalling was blocked by the overexpression of TβRI-KR, significant decreases were observed in *PAI1*, *p21*, and *ITGA5* levels between TβRI-KR- and mock-transfected SK-MEL28R cells (Figure 9G–I). A 20-fold decrease in *PAI1*, a 7-fold decrease in *p21*, and a 34-fold decrease in *ITGA5* levels were observed between TβRI-KR- and mock-transfected SK-MEL28R cells (Figure 9G–I). The decrease in *PAI1* and *p21* levels was expected because they are TGF-β-responsive genes, but the decrease in *ITGA5* suggests that *ITGA5* transcription is controlled by the TGF-β pathway in resistant cells. Also, when resistant cells were grown on fibrinogen, mock-transfected cells showed a 4-fold increase in PAI1 levels, a 3-fold increase in *p21* levels, and a 7-fold increase in *ITGA5* levels compared to parental cells (Figure 9G–I). When mock- and TβRI-KR-transfected resistant cells were compared in the presence of fibrinogen, a 40-fold decrease in *PAI1*, a 16-fold decrease in *p21*, and a 40-fold decrease in *ITGA5* levels were observed (Figure 9G–I). These data also imply that *ITGA5* expression is controlled by the TGF-β pathway in resistant cells, and that the presence of fibrinogen has a cumulative effect by inducing other mechanisms that induce TGF-β signalling.

### 2.9. Gene Expression Analyses to Understand the Effect of Combination Therapy on ITGA5, ITGB3, p21, and PAI1

Acquired resistance was associated with increased *ITGA5*, *PAI1*, and *p21* levels. To determine whether cilengitide and combination therapy could reverse these changes, we analysed *ITGA5*, *ITGB3*, *PAI1*, and *p21* transcription levels in cells treated with vemurafenib, cilengitide monotherapies, and combination therapy at the 2, 4, 6, 8, and 24 h time points.

Sharp decreases were observed in *ITGA5*, *ITGB3*, and *PAI1* transcription levels as a response to cilengitide and combination therapy at different time points (Figure 10). Cilengitide monotherapy significantly decreased *ITGA5* levels at 24 h (Figure 10A), while reducing *ITGB3* levels at 8h (Figure 10C) and *PAI1* at 24 h (Figure 10D). Vemurafenib+cilengitide combination therapy also significantly reduced *ITGA5* levels at 24 h (Figure 10A), while reducing *ITGB3* levels at 8 h (Figure 10C) and *PAI1* at 2 and 24 h (Figure 10D). All of these decreases suggest that cilengitide and combination therapy may act to reverse vemurafenib resistance. No change in *p21* expression was observed at most time points (Figure 10B); the *p21* expression levels seen here might be related to the natural expression in the cell cycle independent of the treatments.

## 3. Discussion

Metastatic melanoma is a deadly skin cancer which has an increasing incidence [23]. Until recently, there were no effective treatments for advanced melanoma. The approval of immunotherapies and inhibitors of BRAFV600E mutant kinase have resulted in some improvements to survival, but not for all patients. BRAFV600E inhibitors, such as vemurafenib, decrease the activity of the MAPK pathway and shrink tumour sizes; however, patients typically develop resistance in around 6 months, resulting in poor prognosis. Therefore, it is important to understand and develop new strategies to overcome vemurafenib resistance.

RGD-binding integrins are involved in signalling and they have important roles in tumour progression and resistance mechanisms. Osteopontin binding to integrin αvβ3 in healthy cells is responsible for proliferation, migration, and apoptosis, but, in breast cancer, promotes cell motility, angiogenesis, metastasis, and tamoxifen resistance [24,25]. Integrin α5β1 promotes cetuximab resistance in head and neck cancer [26]; the overexpression of the integrin β1 subunit induces SRC and AKT signalling and promotes cetuximab resistance in pancreatic cancer [27]. Increased β3, by interacting with galectin-3, thus activating KRAS, RELB, and NF-ΚB pathways, is involved in erlotinib and lapatinib resistance in lung cancer and linsitinib resistance in pancreatic cancer [5,28].

However, the role of these integrins in vemurafenib resistance in metastatic melanoma has not been previously investigated. To explore the involvement of integrins in vemurafenib resistance, the expression of RGD-binding integrins were analysed in resistant cells, and a combination therapy was used to target the increased integrins and disrupt their signalling. Secondly, the TGF-β (canonical, non-canonical), NF-ΚB (canonical, non-canonical), MAPK, PI3K, and PDL1 signalling pathways were investigated. Thirdly, the effects of combination therapy on colony formation and the invasion of resistant cells were analysed.

The RGD binding integrins *ITGA2*, *ITGAIIB*, *ITGA4*, *ITGA5*, *ITGAV*, *ITGB1*, *ITGB3*, *ITGB5*, *ITGB6*, and *ITGB8* were analysed in vemurafenib-resistant A375 and SK-MEL28 cells in the presence and absence of the integrin ligand fibrinogen. Our results revealed that ITGβ3 (in protein expression) and ITGA5 (both in mRNA and protein expression) levels are significantly increased in the development of vemurafenib resistance. Therefore, cilengitide was added to vemurafenib treatment to inhibit the function of α5β1 and αvβ3. We found that there is a synergistic cytotoxic effect in resistant cells as a response to the vemurafenib+cilengitide combination. To analyse the response of resistant cells to the combination, two different tumourigenic features were studied. First, a colony formation assay, which mimics the ability of a single cell to generate a metastatic tumour after reaching a new tissue, and second, the invasive capability using the spheroid Matrigel invasion assay. Significant decreases were found in both colony formation and invasion as a response to combination therapy. In both the colony formation assay and Matrigel invasion assay, cilengitide monotherapy decreased tumourigenic behaviour, while vemurafenib increased it. This paradoxical effect of vemurafenib has been found in previous studies using SK-MEL28-resistant cells [29]. The paradoxical effect of vemurafenib and its reversal by cilengitide can be explained by the upregulation of MMP2 in resistance cells, which induces invasive and migratory properties by modulating αvβ3 location and signalling via fibronectin fragments [30]. The significant effect of vemurafenib+cilengitide combination on colony formation and invasion suggests that this therapy may delay the progression of melanoma in drug-resistant patients.

Several signalling pathways possibly associated with vemurafenib resistance were analysed by looking at the transcription levels of *HRAS*, *MAP2K1*, *MAPK1*, and *MAPK3* (MAPK signalling); of *SMAD4*, *SMAD7*, *PAI1*, and *p21* (TGF-β signalling); of *RELA*, *RELB*, and *CHUK* (NF-ΚB signalling); of *AKT1* (PI3K signalling); and of *PD1* and *PDL1* (PDL1 signalling). The vemurafenib-resistant A375 and SK-MEL28 cells were compared in the presence and absence of the integrin ligand fibrinogen

Significant decreases were observed in *MAPK1*, *MAP2K1*, *IKKα*, *HRAS*, *SMAD4*, *AKT 1*, and *RELA* levels in resistant SK-MEL28 cells, which may indicate a possible decrease in MAPK, TGF-β, and NF-ΚB, but these genes represent pathway elements and not pathway-responsive genes, so these decreases are not directly associated with decreases in signalling. For further analysis, protein experiments are necessary. Also, in the literature, MAPK pathway activity increases due to resistance development [31]. To clarify this finding, p-ERK1/2 levels were analysed, and it was found that p-ERK1/2 levels are higher in resistant cells and, also, both in parental and resistant cells, there is an increase in the presence of fibrinogen, which means that MAPK activation is related to the activation of integrin-associated pathways by ligand binding. However, significant decreases were observed in *MAPK1* and *MAP2K1* transcription levels. Localisation of MAPK proteins were also analysed because it is very common for proteins to change their location due to their activation or inactivation, with their new localisation giving clues about their role in cells. No localisation change was observed in ERK1/2 and p-ERK1/2 levels in resistant cells, but, interestingly, ERK1/2 accumulated in the nucleus in resistant cells in the presence of fibrinogen. ERK1/2 has a cytoplasmic localisation in quiescent cells, but it tends to accumulate in the nucleus in MAPK active cells which have been triggered by ligand binding [32]. Additionally, ERK1/2 does not have a regular nuclear localisation signal, but rather has a nuclear translocation signal, so, by interacting with IMPORTIN7, it uses importin 7 protein as a shuttle and can localise in the nucleus as a quick response to stimulation, mainly to induce the transcription of genes. However, the nuclear localisation of ERK was not observed in resistant cells without the presence of fibrinogen. Our observations might be related to the accumulated induction of the MAPK pathway due to the resistance and the mechanisms induced by the presence of fibrinogen, or both. However, this observation is weak, and more studies need to be performed; inducing and silencing the ERK1/2 and MAPK activity individually at different time points while using different concentrations of fibrinogen may provide a better understanding.

Our results revealed that *ITGB3*, *ITGA5*, *PAI1*, and *p21* levels are significantly increased in vemurafenib-resistant cells. The increase in *PAI1* and *p21* levels as TGF-β-responsive genes indicated that TGF-β is active in resistance; so, by using reporter assays, and with further analysis, it was determined that TGF-β is activated due to acquiring vemurafenib resistance. This is the first study to report on the association between *ITGB3*, *ITGA5*, *PAI1*, and *p21* expression and TGF-β activation in BRAFV600E vemurafenib-resistant melanoma cells and to compare parental and resistant cells in this manner. TGF-β is a known component of drug resistance, and its role in BRAFi resistance has been previously shown. For example, in 2017, Flores et al. showed that *TGFB1*, *TGFBRII*, and *SMAD3*, but not *TGFBRI*, *SMAD2*, and *SMAD4*, transcription levels increased in vemurafenib-resistant SK-MEL28 cells [33]. In our study, to understand the involvement of TGF-β in vemurafenib resistance, the TGF-β pathway was blocked by using a kinase-deficient TGFBRI-expressing plasmid (TBRI-KR). A significant decrease was observed in *PAI1* and *p21* transcription levels, which was expected due to the blockage of TGF-β. Surprisingly, a significant decrease was also observed in the *ITGA5* level. This suggests that *ITGA5* expression is controlled by the TGF-β pathway in vemurafenib-resistant cells. Additionally, the overexpression of constitutively active TGFBRI (TBRI-TD) and TGFBRII activated the TGF-β pathway in resistant cells but not in parental cells, which means that only resistant cells are responsive to the TGF-β pathway. In mock-transfected cells, significantly higher TGF-β activity was observed in resistant cells in comparison with parental cells, which indicates endogenously increased TGF-β activity in resistant cells. On the other hand, the overexpression of SMAD2/3/4 did not activate TGF-β signalling in parental or resistant cells. In TGF-β signalling, SMADs were downstream of the TGF-β receptors and their overexpression was not strictly correlated with TGF-β activity [34]. It has been shown that the overexpression of pSMAD2 and pSMAD3 upregulates TGF-β signalling while SMAD2 and SMAD3 overexpression downregulates it [35], and that there is no linear correlation between the SMAD levels and TGF-β activity, which is consistent with our results. Further studies are required to understand the limits of TGF-β involvement in vemurafenib resistance in melanoma; however, our findings emphasize the role of the TGF-β signalling in BRAF inhibitor resistance and confirm previous findings.

Cilengitide directly inhibits integrin αvβ5, αvβ3, and α5β1 ligand binding [36,37] and decreases TGF-β activity indirectly. It is known that αvβ3 activates LAP-β1, which increases TGF-β1 [38] in human muscle cells, and cilengitide reduces TGF-β activity by inhibiting αvβ3. Our study showed that cilengitide decreased *ITGB3* and *ITGA5* mRNA levels in vemurafenib-resistant cells and reduced the activity of TGF-β, as indicated by the decreased *p21* and *PAI1* levels in cilengitide monotherapy.

Our findings also suggest that TGF-β controls *ITGA5* transcription in vemurafenib-resistant melanoma cells (Figure 9I and Figure 11). It has previously been shown that the activation of the TGF-β pathway controls *ITGA5* expression in human hepatocarcinoma cells [39] and monocytes [40]. TGF-β has also been shown to induce integrin α6, β1, and β4 subunit expression in trastuzumab-resistant breast cancer [41]. These results suggest that TGF-β is a major driver of resistance and works through integrin expression and signalling pathways.

Our findings not only provide insights into the involvement of integrins in vemurafenib resistance, but also highlight the potential of combination therapy with integrin antagonists to delay melanoma progression in drug-resistant patients. Furthermore, our study reveals the role of TGF-β signalling in vemurafenib resistance and its control over *ITGA5* expression. These findings suggest that targeting integrins and TGF-β signalling, specifically *ITGA5*, *ITGB3*, *PAI1*, and *p21*, represents a promising strategy for overcoming drug resistance in melanoma.

## 4. Materials and Methods

### 4.1. Reagents

Vemurafenib, an FDA-approved BRAF inhibitor, was obtained from Stratech [Cambridge, UK]. FBS, sodium pyruvate, L-glutamine, and RPMI were obtained from Gibco [Waltham, MA, USA]. Rabbit polyclonal anti-ITGA5 (AB1928), MTT, DMSO, agarose, and ethidium bromide were obtained from Merck. The RNA-easy kit was obtained from Qiagen [Venlo, The Netherlands]. The iTaq Universal SYBR Green One-Step Kit, Precision Plus Protein™ Standards Clarity™, Western ECL Substrate, and iScrpt-Reverse Transcription Supermix were obtained from Biorad [Hercules, CA, USA]. Lipofectamine 2000 (11668027) was obtained from Invitrogen [Waltham, MA, USA]. The anti-p-ERK 1/2 Antibody (12D4): sc-81492, anti-ERK 1/2 Antibody (MK1): sc-135900, Goat Anti-Rabbit IgG-HRP, Goat Anti-Mouse IgG-HRP, and Mouse monoclonal anti-ITGβ3 (sc-46655) were obtained from Santa Cruz [Santa Cruz, CA, USA]. Goat anti-Mouse IgG (H+L) Highly Cross-Adsorbed Secondary Antibody, and Alexa Fluor 488- A-11029 were obtained from Thermo Fisher [Waltham, MA, USA], and the luciferase assay system (E4030) was obtained from Promega [Madison, WI, USA]. Fibrinogen (Fibrinogen–Bovine plasma-34157) and Mouse anti-β-Actin were obtained from Sigma [Burlington, MA, USA]. SBE-LUC, TβR-II, and pCMV-β-Gal were kindly provided by Talat Nasim, University of Bradford, UK. The p3TP-lux, pARE-lux, and pCMV5B-TGFbeta receptor I K232R were a gift from Joan Massague and Jeff Wrana (Addgene plasmid # 11767, 11768, and 11763 [42,43]).

### 4.2. Cell Culture

A375, SK-MEL2, SK-MEL28, M14, MeWo, and UACC62 cell lines were purchased from ATCC and provided by the Institute of Cancer Therapeutics, University of Bradford. Cell lines were grown in RPMI 1640 media supplemented with 10% FBS, 1% sodium pyruvate, and 2 mM of L-glutamine. A375 and SKMEL28 cells were seeded into six-well plates (2.2 × 10^6^ cells/well) and transfected while seeding with Lipofectamine 2000 according to the manufacturer’s instructions. Fibrinogen was prepared with PBS to a final concentration of 320 ng/mL. Six-well plates were coated with 500 µL/well and 10 cm plates were coated with 1 mL of the fibrinogen solution overnight at +4 °C in sterile conditions. Before use, excess fibrinogen was removed by washing with PBS three times. Cells were seeded as 5 × 10^6^ for 10 cm plates for the real-time PCR and 1 × 10^6^/well for 6-well plates for the real-time and reporter assay experiments.

### 4.3. Cell Viability Assay

Drugs were prepared with DMSO as a 100 mM stock solution. An amount of 200 µL of cells (1 × 10^4^ cells/mL) were seeded in 96-well plates. Cells were treated in the range of 0.01 µM, 0.1 µM, 1 µM, 10 µM, 100 µM, and 1000 µM of drugs. For blank wells, the solvent (DMSO) was used according to the maximum number of dilutions to exclude any effect of the solvent. All of the treatments were performed in triplicate and incubated for 96 h at 37 °C. The media was discarded and the cells were incubated with MTT solution at a final concentration of 0.5 mg/mL. After 4 h, the MTT solution was removed and formazan crystals were resuspended with DMSO. The absorbance was measured using a Multi scan Plus spectrometer at 550 nm (Lab-systems Group, UK), the IC_50_ results were calculated with The Quest Graph™ IC_50_ Calculator [44], and graphs were plotted by Excel (version 2406).

### 4.4. Spheroid Invasion Assay

The hanging-drop method was used to form the spheroids [45]. Previous studies showed that the optimal number of cells to form spheroids was 5000 cells in 100 μL, which was used for these studies. Cells were incubated for 5 days, and then the spheroids were collected and embedded into Matrigel. Each spheroid was mixed with 100 μL of the Matrigel solution and dispensed into 96-well pre-warmed plates (37 °C), and then allowed to solidify for 30 min. Then, the IC_50_ levels of vemurafenib and cilengitide were used for both individual doses and combination treatment. DMSO was used as a control. Spheroids were incubated for 48 h to observe invasion. Data were analysed by ImageJ. The total area of invasion was circumscribed and measured, and then divided by the core spheroid area. Then, the data were normalised using the average of the DMSO-treated spheroids’ invasion area.

### 4.5. Colony Formation Assay

A total of 500 cells had the highest plating efficiency, so they were used for further treatment experiments. The IC_50_ levels of the drugs were used for both individual doses and combination treatment. DMSO was used as a control. Treated cells were incubated for 11 days in fibrinogen-coated and non-coated 6-well plates. At the end of the incubation, cells were washed with PBS and fixed with methanol, and then incubated with 0.1% crystal violet solution. Colonies were counted by eye and the results were analysed using the following formula: Surviving fraction = (Number of colonies formed after treatment/number of cells seeded) × 100 [21].

### 4.6. Quantitative Real-Time PCR

RNA was isolated using a Qiagen RNAeasy kit according to the manufacturer’s instructions. The Biorad-iScript-Reverse Transcription Supermix for RT-qPCR was used for the cDNA synthesis. The PCR was carried out using the iTaq Universal SYBR Green One-Step Kit and CT results were measured using the Applied Bioscience ABI 7500 Real time instrument and 7500 software v1. Amplification: (95 °C for 10 s, 60 °C for 1 min) × 40 cycle; Melting: denaturation—95 °C for 15 s, annealing 60 °C for 1 min, elongation 95 °C for 15 s. Real-time primers (*ITGB1*, *ITGB3*, *ITGB5*, *ITGB6*, *ITGB8*, *ITGA2*, *ITGA2B*, *ITGA4*, *ITGA5*, *ITGAV*, *MAPK1*, *MAPK3*, *MAP2K1*, *RELA*, *RELB*, *p21*, *IKKα*, *HRAS*, *SMAD4*, *SMAD7*, *AKT1*, *GAPDH*, *PAI1*, *PD-1*, and *PD-L1*) were from Sigma KiqStart. Each sample and primer PCR were performed in triplicate as a technical repeat, and experiments were performed three times with different samples. To analyse the RT-qPCR data, the cDNA threshold cycle (Ct) values were normalised by the housekeeping gene *GAPDH*. Data were calculated according to the following formula: Average of technical repeats and ΔC_t_ = C_t_ (average of target gene) − C_t_ (average of housekeeping gene). The 2^−ΔCt^ value was calculated and 2^−ΔΔCt^:2^−ΔCt (sample)^/2^−ΔCt (control)^ was calculated to show the fold changes.

### 4.7. Heatmap Generation

Heatmaps were generated using the Python programming language with libraries including NumPy(1.24.0), Matplotlib(3.6.3), and Seaborn(0.11.2). The provided Python code utilized Google Colab’s environment to create a heatmap visualisation. First, the data and corresponding quantitative values were organised into a numpy array. The Seaborn library’s heatmap function was then employed to visualize this data matrix, with customisations such as handling NaN values using a masking technique. A blue and red colour palette was chosen to distinctly represent positive and negative values. Axes were labelled accordingly, and the resulting heatmap was displayed using Matplotlib.

### 4.8. Reporter Assay

The 3TP-LUX (PAI-1-driven promoter), SBE-LUC (Smad-binding element), and PARE-LUX (activin-responsive element) as TGF-β-responsive reporter plasmids were used to show the activation of TGF-β. TβRI-KR(K232R), TβRI-TD(T204D), and TβR-II as TGF-β receptors were overexpressed as positive and negative controls. Also, pCMV5-Myc-Fast2 was overexpressed with PARE-LUX for its activation. To equalize the amount of plasmid for each well, the total amount was completed with an empty FLAG vector.

Cells were lysed for 24 h after transfection with reporter lysis buffer (Promega, Cat. No. E4030), and the luciferase activity was measured by a luminometer (Fluoroskan ascent FL-Thermo Scientific) immediately after dispensing a luciferase assay system substrate, luciferin, with the luminometer.

The luciferase activity was normalised for transfection efficiency by β-galactosidase activity, quantified by absorption at 405 nm after incubation with ONPG (4 mg/mL)+β-mercaptoethanol + Z buffer, and reaction-terminated by 1 M Na_2_CO_3_ buffer.

The reporter assay data were divided by the β-galactosidase assay results to normalize the transfection efficiency.

### 4.9. Western Blot

Cells were lysed in TNTE (Tris-NaCI-TritonX-100-EDTA) buffer. Samples (20 µg/well) were loaded on 7.5% acrylamide gel in a running buffer which included 10% SDS. The Precision Plus Protein™ Dual Color Standards-Biorad was used as a marker. Proteins were transferred onto PVDF membrane previously soaked in methanol. Transfer was carried out using 1× cold transfer buffer on ice.

PVDF membranes were blocked in 5% skimmed dried milk in TBST. Following blocking, the primary antibodies Mouse monoclonal anti-ITGβ3 (Santa Cruz), Mouse anti-β-Actin (Sigma), and Mouse monoclonal anti-ITGα5 (Santa Cruz) were diluted in 5% skimmed dried milk in TBST at 1/1000 and incubated with the membrane overnight at 4 °C. The secondary antibody, Goat Anti-Mouse IgG-HRP (Santa Cruz), was applied at 1/7500 in 5% skimmed dried milk in TBST and incubated for an hour. Protein bands were detected with a chemiluminescence (Clarity™ Western ECL Substrate-Biorad) reagent and visualised by Biorad chemidoc MP. The Western blot band intensity was quantitated using Image Lab(6.1.0, built 7) (Bio-Rad, Hercules, CA, USA) software and Student’s *t*-test was applied to show the significance.

### 4.10. Immunofluorescence

Sterile glass slides were placed in 6-well plates and covered with 500 µL/well of fibrinogen and PBS. The plates were then left overnight at 4 °C under sterile conditions. Subsequently, 2.2 × 10^6^ cells were seeded per well onto the glass slides in the 6-well plate. Cells were fixed for 10–15 min in methanol, permeabilised with 0.2% Triton X-100 in PBS, blocked for 10 min to 1 h in 10% bovine serum albumin (BSA)–PBS, and incubated for 2 h with primary antibodies, Rabbit polyclonal anti-ITGα5 (Merck AB1928) at 1:500 and Mouse monoclonal anti-ITGβ3 (Santa Cruz sc-46655) at 1:5000, in 10% BSA–PBS, and subsequently incubated for 1 h with secondary antibodies, Goat Anti-Mouse IgG-Alexa Fluor 488 and Goat Anti-Rabbit IgG-Alexa Fluor 488 at 1:100, mounted using DAPI.

### 4.11. Combination Therapy

A375P, A375R, SK-MEL28P, and SK-MEL28R cells were treated with vemurafenib combined with cilengitide in a 1:1 ratio of concentrations, such as 0.25-fold, 0.5-fold, 1-fold, 2-fold, and 4-fold IC_50_ values of the individual drugs (Figure 12), and the effect of the combination was analysed by an MTT assay. The CompuSyn software (version 1.0) was used to calculate the combination index (CI), which indicates the interaction between the drugs. CI > 1.1 represents antagonism; CI < 0.9 represents synergism; CI 0.9–1.1 represents an additive effect [46].

### 4.12. Morphological Analyse of Parental and Resistant Cells

A375, SK-MEL 28, M14, and UACC62 parental and resistant cell lines were cultured and photographed 24 h after seeding without the drug. Using ImageJ, the minor and major axes of the cells were measured. The average length of the major axis divided by the average length of the minor axis was determined for approximately 50 cells from both parental and resistant cell lines. The analysis of the images was repeated for 3 independent experiments, and the averages of the results were analysed by Student’s *t*-test. Significance was shown by * *p* ≤ 0.05, ** *p* ≤ 0.01, *** *p* ≤ 0.001, and *****p* ≤ 0.0001.

### 4.13. Statistical Analysis

Statistical analyses were carried out using a two-way ANOVA variation test and Tukey’s post hoc test in Graphpad, or with a two-tailed Student *t*-test in Excel. Differences were considered significant if * *p* ≤ 0.05, ** *p* ≤ 0.01, ****p* ≤ 0.001, and **** *p* ≤ 0.0001, or non-significant if *p* > 0.05. Error bars represent ± SD of three independent experiments, each conducted in triplicate.

## 5. Conclusions

This study contributes to the understanding of vemurafenib resistance in metastatic melanoma and underscores the importance of integrins and TGF-β signalling in this process. By elucidating the mechanisms underlying resistance development and identifying druggable targets, we hope to provide new therapeutic options that can improve patient outcomes and ultimately overcome resistance to BRAF inhibitors in metastatic melanoma.

## Figures and Tables

**Figure 1 ijms-25-07946-f001:**
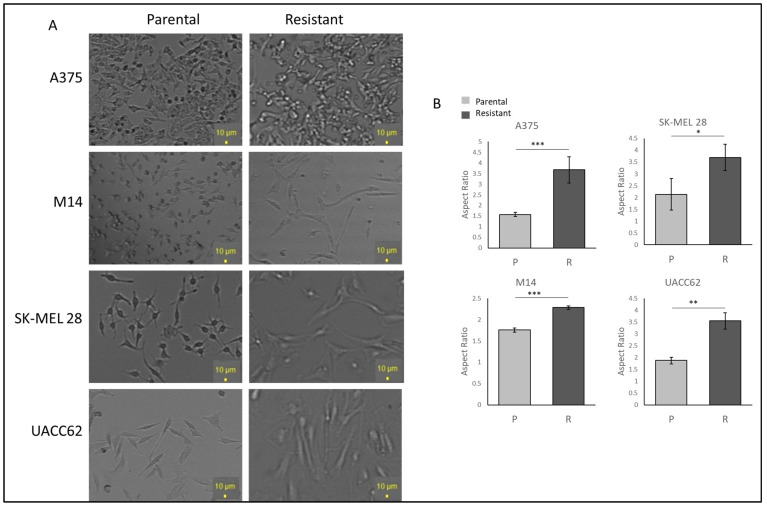
**Vemurafenib-resistant cells show higher IC_50_ values than their parental cells and different morphologies.** (**A**) The morphologies of the parental and resistant cells under light microscopy. Scale bar = 10 µm. (**B**) Aspect ratio comparison of the parental and resistant cells. Image analysis was performed by ImageJ (n = 3 ± SD). * *p* ≤ 0.05, ** *p* ≤ 0.01, *** *p* ≤ 0.001, using Student’s *t*-test.

**Figure 2 ijms-25-07946-f002:**
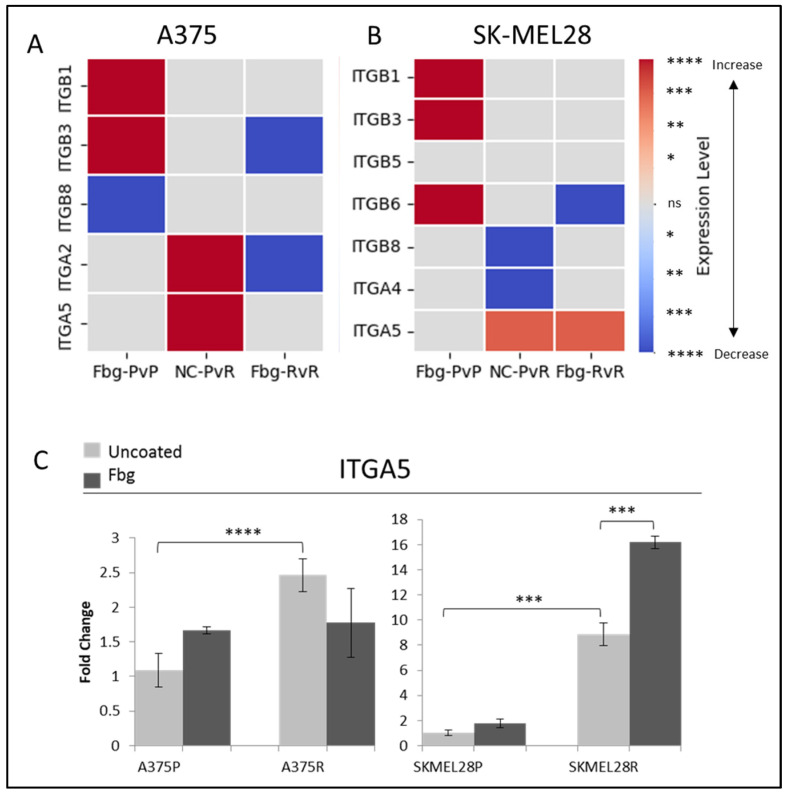
**Vemurafenib resistance increases integrin gene transcription both in A375 and SK-MEL 28 cells.** (**A**) A375 parental cells compared with parental cells grown on Fbg (Fbg-PvP), A375 parental cells compared with resistant cells grown on a non-coated surface (NC-PvR), and A375-resistant cells compared with resistant cells grown on Fbg (Fbg-RvR) 4 h after seeding. (**B**) SK-MEL28 parental cells compared with parental cells grown on Fbg, SK-MEL28 parental cells compared with resistant cells grown on a non-coated surface, and SK-MEL28-resistant cells compared with resistant cells grown on Fbg 6 h after seeding. Integrin genes which did not change significantly under any conditions are not shown. Fbg: fibrinogen-coated, NC: non-coated, P: parental cells, and R: resistant cells. (**C**) *ITGA5* expression in A375 and SK-MEL28 parental and resistant melanoma cell lines in the presence and absence of fibrinogen; the grey bars represent the absence of fibrinogen and the dark-grey bars represent the presence of fibrinogen, (n = 3 ± SD). Statistical analysis was carried out using a two-way ANOVA variation test and Tukey’s post hoc test to show significance. Differences were considered significant, * *p* ≤ 0.05, ** *p* ≤ 0.01, *** *p* ≤ 0.001, and **** *p* ≤ 0.0001 or as non-significant (ns) with *p* > 0.05. Heatmap created with Python.

**Figure 3 ijms-25-07946-f003:**
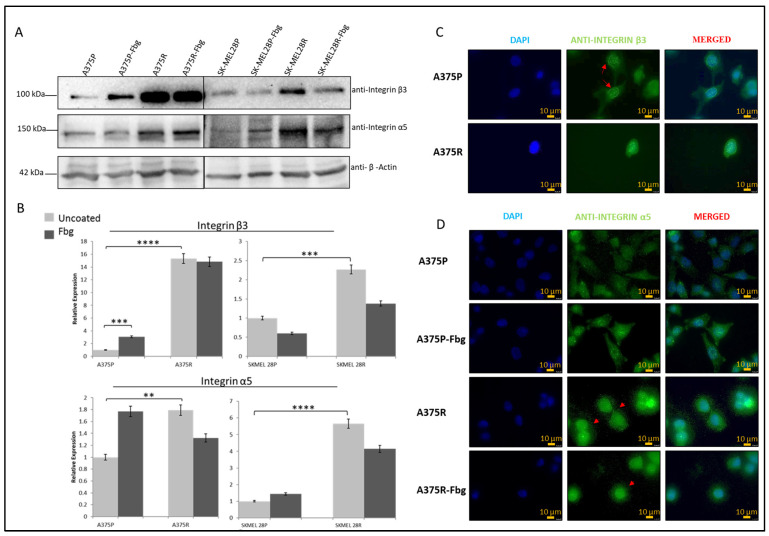
**Vemurafenib resistance increases integrin β3 and α5 protein expression in A375 and SK-MEL28 cells but does not affect the localisation.** (**A**) Protein expression of integrin β3 and α5 subunits in A375 and SK-MEL28 cells in the presence and absence of fibrinogen. (**B**) Quantitation of protein expression normalised with β-actin using ImageLab; the light-grey bars represent the absence of fibrinogen and the dark-grey bars represent the presence of fibrinogen, (n = 3 ± SD). A two-way ANOVA variation test and Tukey’s post hoc test were used to show significance (** *p* ≤ 0.01, *** *p* ≤ 0.001, and **** *p* ≤ 0.0001). (**C**) Expression of β3 in A375 parental and resistant cells: DAPI indicates the nucleus, green is the integrin β3 subunit, and the red arrows show the nucleus membrane localisation. (**D**) Expression of α5 in A375 parental and resistant cells in the presence and absence of fibrinogen: DAPI indicates the nucleus, green is the integrin and integrin α5 protein subunit, and the red arrows indicate α5 in the dot-like structures around the cells. Images are representative of three independent experiments. Scale bar: 10 μm.

**Figure 4 ijms-25-07946-f004:**
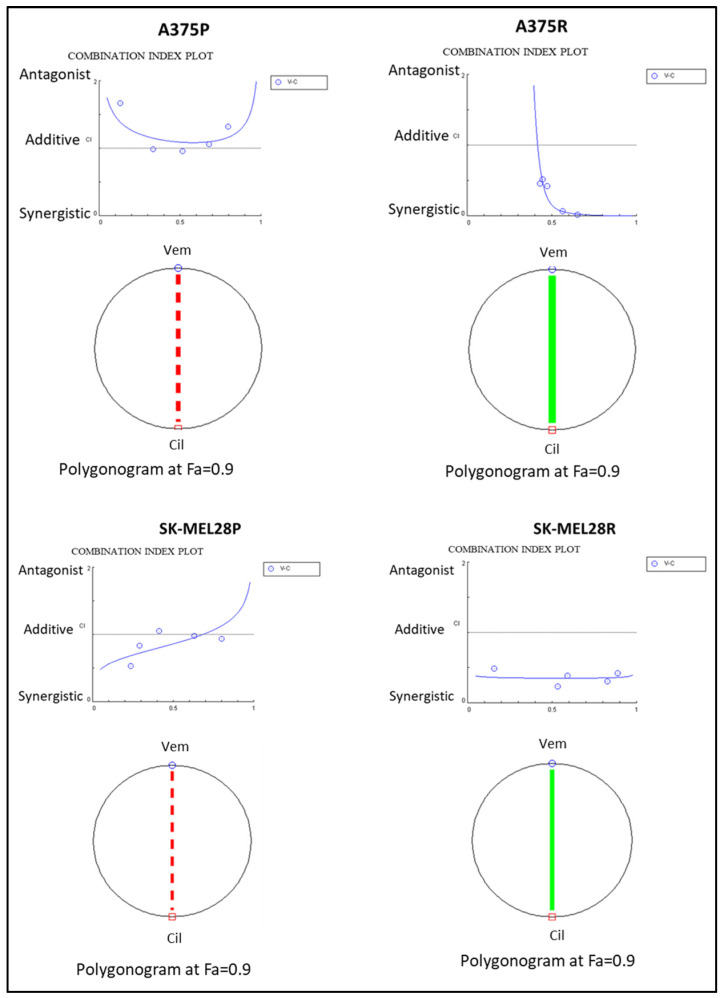
**Vemurafenib+cilengitide combination has a synergistic effect on vemurafenib-resistant A375 and SK-MEL28 cells, but not on parental cells.** The red dashed line indicates moderate antagonism and the green line indicates very strong synergism. Fa: fraction affected; Fa = 0.9 means 90% cell death (n = 3).

**Figure 5 ijms-25-07946-f005:**
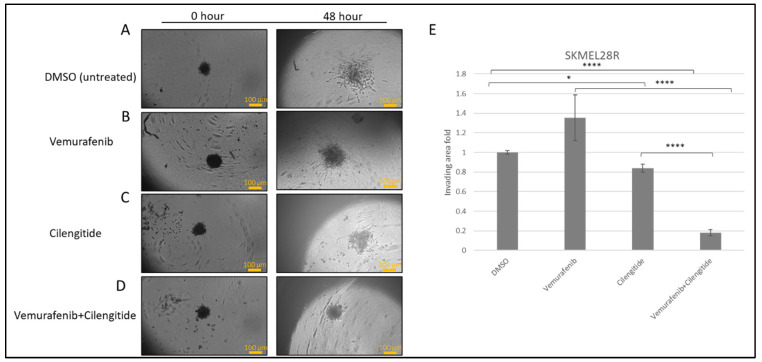
**Combination therapy decreases invasion in SK-MEL28R cells.** Spheroids were formed by SK-MEL28-resistant cells, embedded into Matrigel, and imaged after 48 h. (**A**) Spheroids in Matrigel–control. (**B**) Spheroid invasion in the presence of the IC_50_ concentration of vemurafenib. (**C**) Spheroid invasion in the presence of the IC_50_ concentration of cilengitide. (**D**) Spheroid invasion in the presence of the IC_50_ concentration of vemurafenib+cilengitide. (**E**) The area of invasion was circumscribed and measured by ImageJ, then divided by the core spheroid area and normalised using the average of DMSO-treated spheroids’ invasion area. Scale bar: 100 μm. Statistical analysis was carried out using a two-way ANOVA variation test and Tukey’s post hoc test to show significance. Non-significant differences were determined by *p* > 0.05, * *p* ≤ 0.05, **** *p* ≤ 0.0001, (n = 3 ± SD).

**Figure 6 ijms-25-07946-f006:**
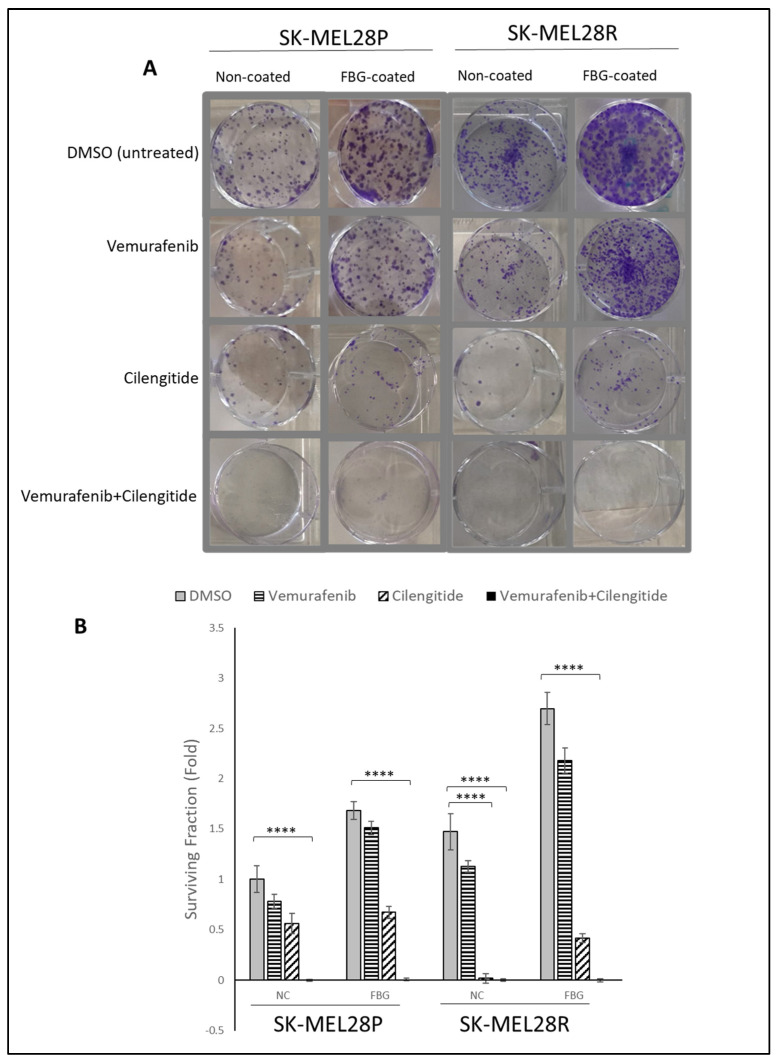
**Combination therapy decreases colony formation in SK-MEL28 parental and resistant cell lines.** (**A**) Images showing the effects of the IC_50_ concentration of vemurafenib, cilengitide monotherapy, and combination on colony formation in the presence and absence of fibrinogen. (**B**) Graphs showing n = 3 ± SD in SK-MEL28 parental and resistant cell colony formation. Statistical analysis was carried out using a two-way ANOVA variation test and Tukey’s post hoc test to show significance. Differences were considered significant, **** *p* ≤ 0.0001 or as non-significant (ns) with *p* > 0.05.

**Figure 7 ijms-25-07946-f007:**
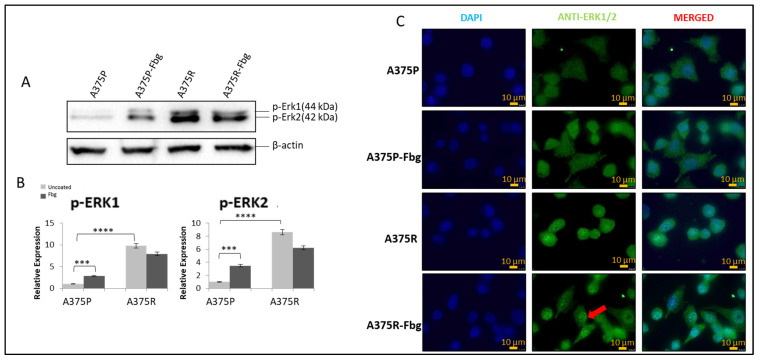
**Vemurafenib resistance induces the MAPK pathway in A375 cells and increases nuclear localisation in the presence of fibrinogen**. (**A**) The p-ERK1 and p-ERK2 expressions in the presence and absence of fibrinogen. (**B**) Protein bands were quantified with ImageJ. Two-way ANOVA variation test and Tukey’s post hoc test were used to show significance. Non-significant (ns) differences were determined by *p* > 0.05, *** *p* ≤ 0.001, and **** *p* ≤ 0.0001, n = 3 ± SD; Western blot results are representative of these experiments. (**C**) Sub-cellular localisation of ERK1/2 in A375 parental and resistant cells in the presence and absence of fibrinogen. DAPI shows the nucleus, green shows the ERK1/2 protein, and red arrows indicate the increased nuclear localisation of ERK1/2 in A375 resistant cells in the presence of fibrinogen. Images are representative of three independent experiments. Scale bar: 10 μm.

**Figure 8 ijms-25-07946-f008:**
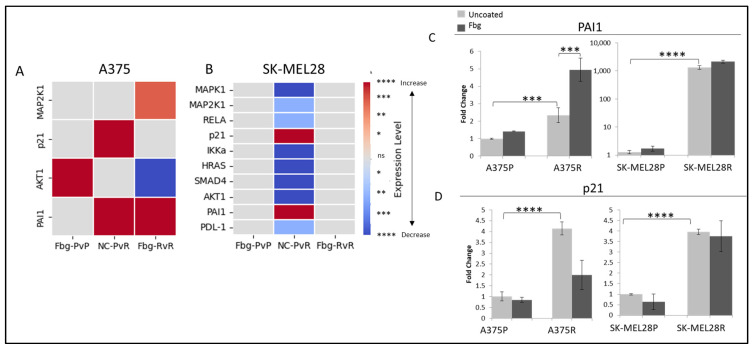
**Vemurafenib resistance increases *PAI1* and *p21* transcription both in A375 and SK-MEL 28 cells**. (**A**) A375 parental cells compared with parental cells grown on Fbg (Fbg-PvP), A375 parental cells compared with resistant cells grown on a non-coated surface (NC-PvR), and A375-resistant cells compared with resistant cells grown on Fbg (Fbg-RvR) 4 h after seeding. (**B**) SK-MEL28 parental cells compared with parental cells grown on Fbg, SK-MEL28 parental cells compared with resistant cells grown on non-coated surface, and SK-MEL28-resistant cells compared with resistant cells grown on Fbg 6 h after seeding. Pathway-associated genes which did not change significantly under any conditions are not shown. Fbg: fibrinogen-coated, NC: non-coated, P: parental cells, and R: resistant cells. (**C**) *PAI1* expression. (**D**) *p21* expression in A375 and SK-MEL28 parental and resistant melanoma cell lines in the presence and absence of fibrinogen; the grey bars represent the absence of fibrinogen and the dark-grey bars represent the presence of fibrinogen, n = 3 ± SD. Statistical analysis was carried out using a two-way ANOVA variation test and Tukey’s post hoc test to show significance. Differences were considered significant, * *p* ≤ 0.05, ** *p* ≤ 0.01, *** *p* ≤ 0.001, and **** *p* ≤ 0.0001 or as non-significant (ns) with *p* > 0.05.

**Figure 9 ijms-25-07946-f009:**
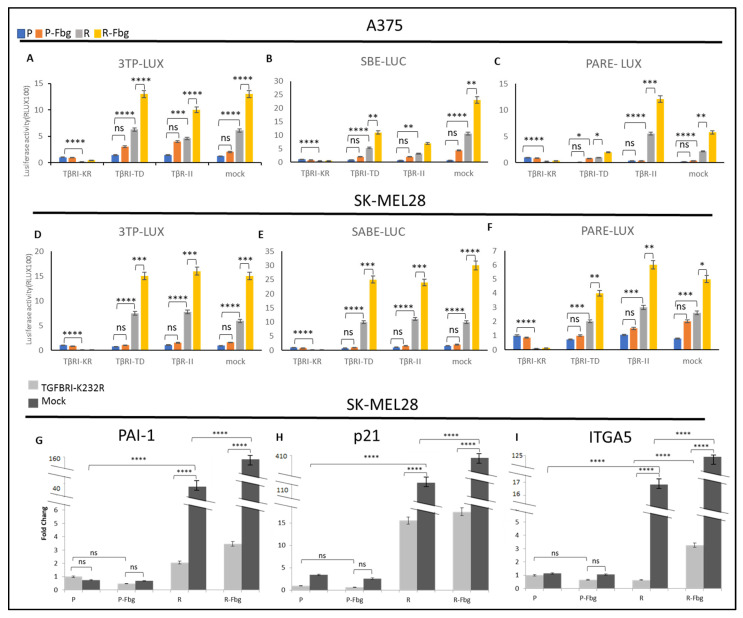
**Vemurafenib resistance induces TGF-β activation and TGF-β controls *ITGA5* transcription in vemurafenib-resistant melanoma cells.** A375 parental and resistant cells’ (**A**) response of 3TP-Luc to the overexpression of TβRI-KR, TβRI-TD, TβR-II, and mock. (**B**) The response of SABE-Luc to the overexpression of TβRI-KR, TβRI-TD, TβR-II, and mock. (**C**) The response of PARE-LUX to the overexpression of TβRI-KR, TβRI-TD, TβR-II, and mock. SK-MEL28 parental and resistant cells’ (**D**) response of 3TP-Luc to the overexpression of TβRI-KR, TβRI-TD, TβR-II, and mock. (**E**) The response of SABE-Luc to the overexpression of TβRI-KR, TβRI-TD, TβR-II, and mock. (**F**) The response of PARE-LUX to the overexpression of TβRI-KR, TβRI-TD, TβR-II, and mock. Blue bar: parental; Orange bar: parental in the presence of fibrinogen; Grey bar: resistant; Yellow bar: resistant in the presence of fibrinogen. Cells transfected with TβRI-KR: TGFB receptor 1 kinase-deficient, TβRI-TD constitutively active receptor 1, TβR-II, mock: transfected with empty plasmid as the control. Cells transfected with TβRI-KR and mock: (**G**) *PAI1*, (**H**) *P21*, and (**I**) *ITGA5* levels determined by real-time PCR. Dark-grey bars indicate the mock transfected with empty plasmid as the control. Light-grey bars indicate transfection by TβRI-KR (K232R) to prevent signal transmission for TGF-β. Fbg: presence of fibrinogen, P: parental, and R: resistant. (n = 3 ± SD). Two-way ANOVA variation test and Tukey’s post hoc test were used to show significance. Differences were considered significant, * *p* ≤ 0.05, ** *p* ≤ 0.01, *** *p* ≤ 0.001, and **** *p* ≤ 0.0001 or as non-significant (ns) with *p* > 0.05.

**Figure 10 ijms-25-07946-f010:**
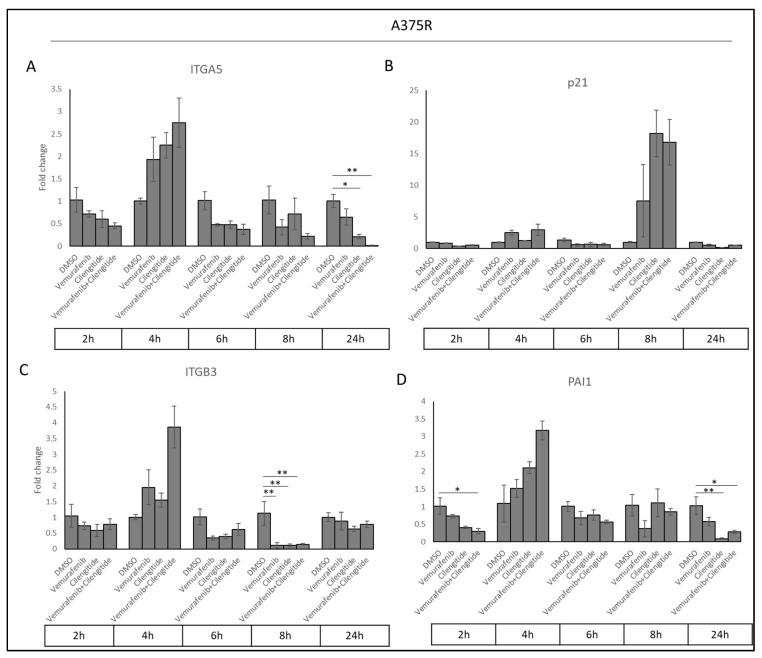
**Combination therapy decreases *ITGA5*, *ITGB3*, and *PAI1* levels in a time-dependent manner.** (**A**) *ITGA5*, (**B**) *p21*, (**C**) *ITGB3*, and (**D**) *PAI1* expression changes in response to treatment with vemurafenib, cilengitide, and combination therapy in A375R cells at time points of 2, 4, 6, 8, 24 h. * *p* ≤ 0.05, ** *p* ≤ 0.01, (n = 3 ± SD).

**Figure 11 ijms-25-07946-f011:**
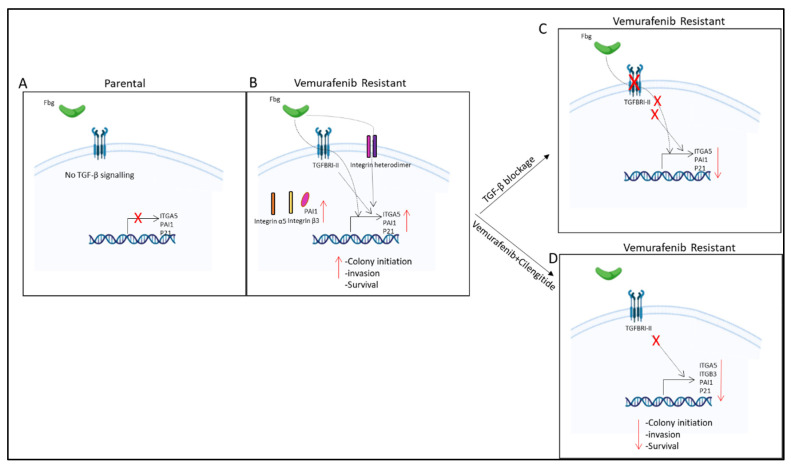
**Summary of results.** (**A**) No TGF-β signalling observed in parental cells. (**B**) TGF-β pathway is active and fibrinogen increases *ITGA5*, *PAI1*, and *p21* gene expression; PAI1, integrin α5, and β3 protein levels; and invasion, survival, and colony-initiating ability in resistant cells. (**C**) TGF-β blockage by TGBRI-KR decreased *ITGA5*, *p21*, and *PAI1* in vemurafenib-resistant cells. (**D**) Vemurafenib+cilengitide combination decreased *ITGA5*, *p21*, *PAI1*, and *ITGB3* levels and colony-initiating ability, invasion, and survival in vemurafenib-resistant cells.

**Figure 12 ijms-25-07946-f012:**
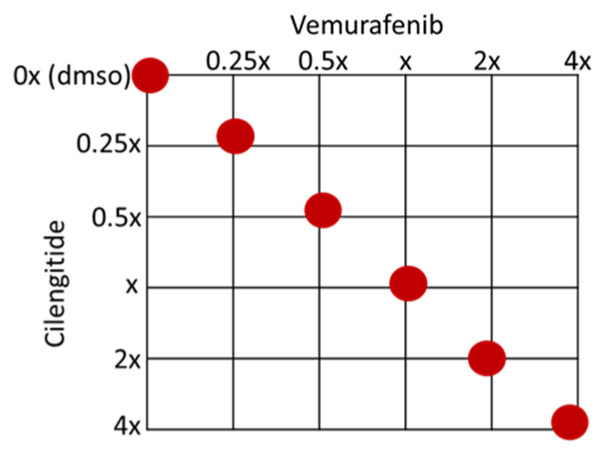
Checkerboard of the vemurafenib+cilengitide combination therapy experiment. Red dots show the concentrations tested as multiples of IC_50_ values.

**Table 1 ijms-25-07946-t001:** BRAFV600E mutant melanoma cells are more sensitive to vemurafenib than BRAF WT melanoma cells (n = 3 ± SD).

	Cell Line	IC_50_ (µM)
BRAFV600E mutant	UACC62	0.14 ± 0.5
	A375	0.17 ± 0.6
	SK-MEL28	0.19 ± 0.6
	M14	0.45 ± 0.1
BRAF WT	MeWo	22.35 ± 3.2
	SK-MEL2	21.6 ± 2.1

**Table 2 ijms-25-07946-t002:** IC_50_ of parental (P) and resistant (R) cells of A375, M14, SK-MEL28, and UACC62 after 4 months of continuous vemurafenib treatment.

Cell Lines	IC_50_ (µM)-P	IC_50_ (µM)-R
A375	0.6 ± 0.1	13.2 ± 2.2
M14	0.7 ± 0.1	8.8 ± 1.2
SK-MEL28	0.2 ± 0.05	13.2 ± 1.2
UACC62	0.2 ± 0.05	17.2 ± 1.5

**Table 3 ijms-25-07946-t003:** **A375 parental–resistant and SK-MEL28 parental–resistant cell line responses to cilengitide**. MTT results of A375P, A375R, SK-MEL28P, and SK-MEL28R cell lines (n = 3 ± SD).

Cell Line	IC_50_ (µM)
A375P	0.5 ± 0.1
A375R	0.5 ± 0.06
SK-MEL28P	1.5 ± 0.2
SK-MEL28R	0.6 ± 0.1

**Table 4 ijms-25-07946-t004:** Vemurafenib+cilengitide combination has a synergistic effect on vemurafenib-resistant A375 and SK-MEL28 cells, but not parental cells. Data analysed using Compusyn 1.0 (n = 3).

Cell Line	Drug Combo CI Value	Description of Effect
A375P	1.35	Moderate antagonism
A375R	0.00	Very strong synergism
SK-MEL28P	1.33	Moderate antagonism
SK-MEL28R	0.36	Synergism

## Data Availability

The datasets and materials used and/or analysed during the current study are available from the corresponding author upon reasonable request.

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
