# Peer review of "Overcoming Vemurafenib Resistance in Metastatic Melanoma: Targeting Integrins to Improve Treatment Efficacy"

_ijms, 2024, doi:10.3390/ijms25147946_

Round 1
Reviewer 1 Report
Comments and Suggestions for Authors
In this manuscript, Boz et al identified the key molecular factors that implicated vemurafenib resistance in metastatic melanoma. They performed several molecular cell biology experiments, biochemistry assays and qRT-PCR to determine the biological role of integrins and TGF-β signalling in vemurafenib resistance in melanoma and to explore the potential of combining vemurafenib with cilengitide in treating melanoma. They identified PAI1 and p21 transcription, and ITGA5 mediated by TGF-β pathway influenced vemurafenib resistant melanoma cells. The manuscript lacks clarity and the figures require further work.
Specific comments:
1. In the Methods section, what software (e.g. SPSS or PRISM etc.) was used to perform the statistical analyses? The authors should add section 2.11 for statistically analyses.
2. In the Introduction, the authors can further elaborate how integrin-mediated signalling can impact the tumour microenvironment. The authors should also discuss some trial studies related to BRAF inhibitors, and how pre-clinical work can help to improve patient outcomes. This will strengthen the importance of their work.
3. In Figure 1, the plots can be improved. Although the authors made the attempt to keep the y-axis consistent by adding one decimal places to all the plots, it is rather confusing and hard to read the plot. Can the authors please remove the decimal place and be consistent with the intervals chosen for the plot if appropriate?
4. In Figure 2, can the authors re-do the plots? To provide clarity, there should be only 2 subfigures; Figure 2A and 2B. Figure 2A should include the heatmap only, and Figure 2B which is Figure 2G. Can the authors used ggplot R package or PRISM to plot the heatmap. The colour chosen for the heatmap are difficult to interpret. This also applied to Figure 8.
5. In Line 302-303, the authors stated “…..there is no correlation with the re-302 sistance to vemurafenib”. Did the authors perform correlation testing (e.g. spearman or pearson)? If so, it was not clear in the manuscript.
6. In the Discussion section, can the authors discuss about targeted therapy (i.e. vemurafenib, BRAF inhibitor) combined with immunotherapy? Will that help to overcome resistance? Clinical trial studies include NeoTrio and DREAMseq.
7. Can the authors follow the nomenclature and format for genes? It is inconsistent in the manuscript. All gene names/symbols should be italicised.
Reviewer 2 Report
Comments and Suggestions for Authors
This manuscript, entitled "Overcoming Vemurafenib Resistance in Metastatic Melanoma: Targeting Integrins to Improve Treatment Efficacy," demonstrated that integrin a 5 and TGF-β signaling are associated with Vemurafenib resistance in metastatic melanoma, and a combination of targeting integrins by cilengitide with Vemurafenib could improve the efficacy of BRAF inhibitor treatment in melanoma. The study is the most important and of great interest. However, related studies have been reported.
There are some specific comments:
First, questionable for methods:
1) In Method section 2.2 cell culture, the authors seeded 2.2 x 10e6/well cells in the six-well plate at a 4—60% confluence. To our knowledge, this is too many cells for A375 and SK-mel28 appear 100% confluent after several hours.
2) In section 2.10, combination therapy, a better assay design should be used in the checkerboard design.
Second, questionable for results:
1) Fig 1C line 215 should be Fig 1B.
2) Fig 1B is unclear; they can not see the cell morphology.
3) Fig 2 A-F should be better presented as a column figure.
4) Fig 2G was not mentioned in the manuscript.
5) Fig 3C is not clear too.
6) Could the authors explain why A375R was not sensitive to cilengitide?
7) It is hard to see the difference between groups A and D.
8) The difference between the parental and resistant cells for Vemurafenib in Fig 6A is not seen.
9) Do the authors find the difference of TGFβ between the parental and resistant cells?
Round 2
Reviewer 2 Report
Comments and Suggestions for Authors
Thank the authors responsible for all the questions. However, the cell images all look blurred and need better focus.
Author Response
The cell images all look blurred and need better focus.
Thank you for your valuable feedback. We have sharpened the immunofluorescence images as suggested. However, we were using an older fluorescence microscope for these experiments, which may explain the limitations in image quality. Unfortunately, we do not have the opportunity to retake the images at this time. We appreciate your understanding.